# Entomopathogenic Fungi as Alternatives to Chemical Acaricides: Challenges, Opportunities and Prospects for Sustainable Tick Control

**DOI:** 10.3390/insects15121017

**Published:** 2024-12-22

**Authors:** Mahvish Rajput, Muhammad Sohail Sajid, Nasir Ahmed Rajput, David Robert George, Muhammad Usman, Muhammad Zeeshan, Owais Iqbal, Bachal Bhutto, Muhammad Atiq, Hafiz Muhammad Rizwan, Ian Kirimi Daniel, Olivier Andre Sparagano

**Affiliations:** 1Department of Parasitology, University of Agriculture Faisalabad, Faisalabad 38000, Pakistan; drmehvish.aslam@gmail.com (M.R.); dr.zee474@gmail.com (M.Z.); 2Department of Plant Pathology, University of Agriculture Faisalabad, Faisalabad 38000, Pakistan; nasirrajput81@gmail.com (N.A.R.); mani.gee3714@gmail.com (M.U.); dratiqpp@gmail.com (M.A.); 3Reader in Precision Agronomy, School of Natural and Environmental Sciences, Newcastle University, Newcastle upon Tyne NE1 7RU, UK; 4Riphah College of Veterinary Science, Riphah International University, Raiwand Road, Lahore 54000, Pakistan; 5State Key Laboratory for Conversation and Utilization of Bio-Resource in Yunnan, Yunnan Agricultural University, Kunming 650000, China; owais.iqbal.918@gmail.com; 6Department of Veterinary Parasitology, Sindh Agriculture University, Tandojam 70060, Pakistan; bbhutto@sau.edu.pk; 7Section of Parasitology, Department of Pathobiology, KBCMA College of Veterinary and Animal Science, Narowal, Sub Campus UVAS, Lahore 54000, Pakistan; hm.rizwan@uvas.edu.pk; 8Department of Veterinary Pathobiology, School of Veterinary Medicine & Biomedical Sciences, Texas A&M University, College Station, TX 77843, USA; ian.daniel@tamu.edu; 9UK Management College, College House Campus, Stanley St., Openshaw, Manchester M11 1LE, UK

**Keywords:** *Beauveria bassiana*, biological control, *Entomopathogenic fungi*, *Metarhizium anisopliae*, ticks

## Abstract

Ticks are one of the most problematic parasitic pests, world-wide. Infesting livestock, people, and their pets, ticks cause direct negative effects on their hosts through blood-feeding whilst also spreading significant diseases of veterinary and medical concern (e.g., Lyme disease). Controlling ticks through conventional chemical approaches is hampered by challenges associated with product performance, availability, and environmental safety, yet effective tick management is vital—particularly as populations of some species may be expanding because of climate change. To control several pest species in a more sustainable manner, researchers have increasingly been exploring the use of beneficial biological organisms as “biopesticides”, including entomopathogenic fungi. These fungi cause diseases in insects and other invertebrates in the natural environment, and many have shown promise for development as biopesticides against a range of pest species, ticks included. This review considers the potential of these beneficial fungi in controlling ticks, providing examples of their effective use against these parasitic pests from countries around the world. Details on the mode of action of entomopathogenic fungi against ticks, advantages and challenges to their use, and potential applications and prospects for their future practical development as biopesticides are also included.

## 1. Introduction

Ticks are obligatory blood feeders that transmit a diverse array of protozoan, bacterial, and viral pathogens of zoonotic importance. Ticks and mosquitoes have been reported as the main vectors of human and veterinary pathogens globally, although ticks are known to surpass all other arthropods in the variety of infectious agents they transmit [1,2]. Besides their role as biological and mechanical disease vectors, ticks have various additional direct impacts on the health and wellbeing of affected hosts, including blood loss, alopecia, fatal paralysis (where some ticks are also able to inject toxins), and exsanguination [3], with impacts affecting humans, livestock and wild animals alike. Infestations vary across this wide host range, dependent on host preference and agro-ecological location, with the availability of suitable habitats being affected by our changing climate.

Even though climate change has been reported as one of the main drivers of biodiversity loss, some species are able to respond by extending their geographical boundaries. Several tick species survive only in specific climatic conditions, for example, with changes in climate directly impacting their occurrence, while others are capable of adapting to a wide range of climatic conditions [4]. Additionally, generalist feeders and species known to utilize alternative hosts, such as the deer tick *Ixodes scapularis*, the cattle fever tick *Rhipicephalus microplus*, and the sheep tick *Ixodes ricinus*, are more likely to shift their expansion ranges as opposed to strictly host-specific ticks [5,6,7]. As in the tropics, many temperate regions are now characterized by hot and humid conditions that allow a more diverse tick population to establish, with the bulk of the literature on climate change implications on tick populations coming from North America or Europe [4].

The most studied ticks tend to be the most important from a medical and veterinary perspective [4], whereas most significant species globally in terms of abundance, distribution, and contribution to tick borne diseases (TBDs) include the brown dog tick *Rhipicephalus sanguineus* (the most prevalent tick globally), *I. ricinus* (dominating in Europe), *Dermacentor reticulatus* and *Ixodes persulcatus* (found in Siberia and more widely in Asia), the deer tick *I. scapularis* (occurring in most areas of North America and being a key vector of Lyme disease), *Ixodes cookei* (being present in most regions of the Quebec province in Canada and found throughout Northeastern North America), the Rocky Mountain wood tick *Dermacentor andersoni* (found in North, Central, and South America), *Amblyomma* ticks (occurring in South America, especially *Amblyomma neumanni*, *Amblyomma cajennense*, *Amblyomma trieste*, *Amblyomma ovale* and *Amblyomma aureolatum*) and the paralysis tick *Ixodes holocyclus* (known to occur in Queensland, Australia) [4,8,9].

Where ticks are present around the world, a major constraint for livestock animals and humans is the burden imposed by the TBDs they spread [10,11]. Both *Ixodid* and *Argasid* ticks are implicated in the transmission of dangerous pathogens, [12], where the *Ixodid* tick fauna consists of over 700 species worldwide, divided into two groups: the Prostriata (including the *Ixodes* genus) comprised of over 200 species, and the Metastriata (other whitemolecular markers have been studied and employed alongside conventional approaches, species identification is still problematic (especially for cryptic species) and is often challenged for many tick species, leading to controversy in nomenclature in the literature [13].

Tick-borne diseases are increasingly being identified as causes of human disease in many countries. The expansion of vector-borne diseases into diverse regions results from a complex interplay of multiple factors, including climate change, shifting host habitats, land use alterations, global trade, and animal movement [14,15,16]. Ticks also constitute a considerable hindrance to animal production, presenting significant risks that include morbidity and mortality through paralysis and anemia, welfare problems such as dermatitis, and associated economic impacts to livestock farming operations in countries where they occur [1,17]. The continued and sustainable production of milk and meat products from these animals is already a global concern when it comes to food security [18], with tick blood feeding from and pathogen transmission to these hosts adding to concerns. Ticks are responsible for the transmission of major destructive diseases in livestock, for example, Lyme disease and tick-borne encephalitis [1], and jeopardize food safety as a result [19]. These factors, coupled with the common occurrence of ticks developing resistance to acaricides presently used in their control, necessitate the need for novel control approaches coupled with integrated surveillance and pest management.

Synthetic acaricides have remained the gold standard for the control of ticks in recent decades despite delivering detrimental unintended consequences such as environmental pollution, human health risks, food contamination, and the development of resistance in the target pest population [20]. Numerous microorganisms also display pest control potential, however, including nematodes, fungi, and crystalliferous bacteria, all of which have been used to control a range of arthropod pests, including mosquitoes, and some of which have also been used against ticks. Biological control may also be delivered by larger species, where the parasitoid wasp *Ixodiphagus hookeri* was used for tick control in the 20th century [21], and predators such as birds, ants, rodents, shrews, and spiders have similarly been deployed to the same end. Notably, poultry can play a key role in removing ticks directly from infested animals, as well as from livestock housing [22].

From the above naturally derived solutions, the development of EPFs has demonstrated particularly promising potential as an “alternative” tick control method [23]. Working in this area, researchers have demonstrated that several factors play a critical role in the effectiveness of EPFs against these pests, including (1) tick innate responses to infection with EPFs; (2) mechanisms that potentiate the virulence of EPFs, including the impacts of toxic components and EPFs metabolites on target vectors; (3) propensity for mass production of EPFs strains adapted to broad spectrum climatic conditions across the wide geographic ranges that various tick vectors occur; and (4) identification and selection of rigorously virulent EPFs isolates [24].

Operator safety when deploying any pest control product is of utmost importance when implementing any biological control method, and EPFs have demonstrated a favorable safety profile, adding to their appeal as a reliable and sustainable solution for tick control [25]. Similar to other pest control agents, their use is subject to rigorous experimentation on target and non-target toxicity, risk assessments, and regulatory frameworks to ensure their safe and responsible application. These measures help minimize any potential risks associated with their deployment and guarantee that EPFs are applied in a manner that aligns with environmental and safety standards [26].

Previous reports have shown that treatments with EPFs exhibit anti-tick activity by inducing myco-acaricidal effects and inhibiting host reproduction [27]. According to White et al. [28], 17 fungal species have been isolated from ticks, with a few of these species being acari-specific and others being more generalist in their host range, infecting both insects and acari. Such species are especially interesting for their potential to target multiple pests, where two such generalist EPFs have been recently applied in tick biological control, namely *M. anisoplae* sensu lato and *B. bassiana* [28,29].

*M. anisoplae* is a prevalent entomopathogen that thrives on the cuticle of its host. It belongs to the Ascomycota phylum and the Clavicipitaceae family, with review and reclassification of the *Metarhizium* species complex recently undertaken [30]. Although this fungus has already undergone extensive examination and development as a biopesticide, studies suggest that further improvements could be realized through genetic engineering of *M. anisopliae* and formulation development to enhance application and effectiveness [31,32]. *Beauveria bassiana* is also widely used as a biopesticide, being an ascomycete arthropod pathogen belonging to the *Cordycipitaceae* family and the *Hypocreales* order [33]. Strains of *B. bassiana* have been effectively deployed against eggs, larvae, nymphs, and adults of the cayenne tick, *A. cajennense s.l.* and brown dog tick, and *R. sanguineus* s.l., among others [34,35,36,37]. A recent study has also compared the effectiveness of spray applications of *M. anisopliae*, *Metarhizium brunneum*, and *B. bassiana* against *Dermacentor albipictus* larvae, finding that *M. anisopliae* and *M. brunneum* isolates provided 74–99% control of unfed questing ticks and engorged larvae, while *B. bassiana* at similar concentrations delivered 30–64% control [38]. Authors elsewhere have further confirmed that *M. brunneum* is an effective anti-tick fungus under semi-field conditions [39], supporting its potential for wider commercial development and registration against ticks [40]. Similarly, Alonso-Díaz et al. [41] reported high mortality in various life stages of several tick species when exposed to *M. anisopliae* and *M. anisopliae* s. l. [18].

In addition to offering stand-alone solutions for tick control, Mesquita et al. [29] reported that EPFs disrupt the *R. microplus* gut microbiota, positively influencing tick susceptibility to acaricides. The tick gut microbiome is being increasingly targeted in medical and veterinary medicine following the realization that the composition of the microbiome could influence the vectorial capacity and biology of ticks [38,42,43,44]. Using 16S rRNA amplicon sequencing, researchers have gained deeper insights into the microbiome of different tick tissues [45,46] to support this area of research. Interactions between the tick and its microbiota regulate the tick peritrophic matrix and enhance tick epithelial integrity, vectorial capacity, and the pathogen transmission process, highlighting a potential target for anti-tick interventions [43,47]. Tick microbiota and bacterial symbionts that modulate the tick immune responses are, therefore, becoming new targets in tick control approaches [38].

Collectively, the body of work cited above provides strong evidence that ticks can succumb to EPF-mediated biological control methods, with the potential for combined use with other products. Further developing such integrated and sustainable tick control strategies could deliver significant gains for enhanced animal and human health.

## 2. The Significance of EPFs in Tick Control

The term “entomopathogen” encompasses a range of microorganisms that include parasites, bacteria, viruses, or fungi that are capable of infecting insects and other terrestrial arthropods, including mites, ticks, and spiders. These entomopathogens rely on a heterotrophic metabolism, which necessitates a life dependent on a host [48]. *Bacillus thuringiensis*, for example, is a rod-shaped and spore-forming bacterium that has been widely deployed to control defoliating insects in several sectors, particularly forestry [49]. The primary focus of the current review, however, is on EPFs, where there already exists a wide range of EPF-based biopesticides available in the market, most of which are derived from organisms belonging to the genera *Beauveria*, *Metarhizium*, *Akanthomyces*, and *Cordyceps* (all of which belong to the *Ascomycota* in the Order *Hypocreales*). These fungi have the ability to target a range of pests, thereby effectively combating various insect/acari-borne pathogens, including those carried by ticks [50]. To date, over a thousand EPF species have been identified as exhibiting a wide range of strategies and adaptations to successfully invade, reproduce within, and ultimately harm their arthropod hosts. This vast number of EPF species highlights the significant role they play in regulating insect populations in natural systems and their potential for application in pest management strategies [51]. Entomopathogenic Hypocreales are particularly well-specialized pathogens with a range of adaptions that have evolved to effectively infect insects and mites, including the ability to circumvent the host’s immune system defenses and the production of enzymes and degrading substances that can break down the host’s cuticle. As noted above, previous studies have already highlighted the potential of EPFs as a promising alternative to chemical pesticides for tick control, either in field or laboratory settings, with their use aligning with the growing demand for eco-friendly pest management practices [52].

One of the most notable advantages of EPFs is that they naturally exist in the environment and have a specific affinity for targeting arthropods such as ticks [18,53], thus minimizing impacts on higher non-target organisms and the environment [24]. Being naturally occurring, these fungi pose little discernable threat to water sources, soil quality, or air pollution, making them a sustainable and ecologically responsible solution to managing pests [54], ticks included [55]. Moreover, these fungi possess the ability to adapt and evolve with their host populations, reducing the likelihood of pests developing resistance to them [56].

In addition to implementing robust regulatory measures to approve EPF-based pest control products, efforts to raise public awareness and education on the benefits and safety of EPFs can play a crucial role in fostering acceptance and understanding of these sustainable solutions for tick control. By promoting knowledge and understanding, we can encourage the responsible use of EPFs as an effective and environmentally friendly method for managing tick populations [57]. Ultimately, the use of EPFs for tick control aligns with sustainable and eco-friendly practices, fostering a healthier environment for both humans and wildlife and aligning with the One Health principles.

## 3. The Use of EPFs in Tick Control

Entomopathogenic fungi are already widely utilized in some areas of pest management, with certain species such as *B. bassiana* and *M. anisopliae* emerging as “classic” examples of biopesticide candidates in particular geographic regions, offering alternatives to chemicals pesticides where these are neither economically nor ecologically sustainable [18]. Entomopathogenic fungi are nevertheless highly diverse, being heterotrophic, unicellular/multicellular, producing both sexual and asexual spores, and having wide global distributions [58]. They may also exert both direct and indirect effects on their hosts that can be leveraged in pest management, including in ticks. *Metarhizium* spp., for example, have been shown to deliver sub-lethal effects on ticks under field conditions by affecting their feeding behavior [59,60,61].

Whether through direct or indirect activity, EPFs show the potential to control ticks across all pest life stages, including eggs, larvae, nymphs, adults, and engorged females [62,63], where this control is perhaps unsurprisingly reported to also reduce the incidence of the diseases that these ticks spread [64]. To collate examples to support this potential, the current review adopted an unstructured approach, utilizing general search terms (e.g., “entomopathogenic fungus” and “biological control of ticks”) to identify the relevant literature. The resulting body of work identified confirms that studies to demonstrate this potential have been undertaken globally in a range of tick species, using a range of EPFs, in both laboratory and field settings, as summarized in Table 1 and Table 2.

In all, 39 laboratory and 29 field studies were identified that considered the effects of EPFs on a total of 15 different tick species between the years 1998 and 2024. Across these studies, nine species of fungi had been investigated, with the majority of studies focusing on *M. anisopliae* and *B. bassiana*, and most of this work being undertaken in Mexico, followed by Brazil and the USA. These studies reveal a wide range of outcomes regarding the use of numerous EPFs against varied tick species and life stages, which can be attributed to differences in study scope (e.g., target tick species, EPF strains), geographical locations, and variations in biotic and abiotic parameters. For instance, the choice of acaropathogen and its formulation can play a crucial role in tick control, where in work by Kirkland et al. [65], both *M. anisopliae* and *B. bassiana* were effective against *R. microplus* ticks, delivering a decrease in egg hatchability (EH), an increase in egg incubation period (EIP), and an increase in egg hatchability period (EHP). Similarly, [66] reported a reduction in egg hatchability of *Rhipicephalus appendiculatus* and *Amblyomma variegatum* after exposure to *B. bassiana* and *M. anisopliae*, but with both EPFs displaying variation in their efficacy dependent on their formulation (i.e., whether this was oil-based or aqueous). Differences may have also been attributed to variations in EPF strain virulence or environmental factors, where variations in tick species and their susceptibility to specific EPFs were evident [67,68,69]. Pirali-Kheirabadi et al. [67] reported a decrease in EH for *A. cajennense* when exposed to *M. anisopliae*, for example, whilst [69] did not find significant differences in EH for the same tick species when treated with *B. bassiana*. Onofre et al. [69] further showed varied results for *Anocentor nitens* ticks, with EH remaining largely unchanged when exposed to *B. bassiana*, but a decrease was seen in both EHP and egg hatchability index (EHI). It is important to consider such differences when designing tick control strategies, as the efficacy of EPFs may vary depending on any of these factors, as well as interactions between them.

**Table 1 insects-15-01017-t001:** Examples from the literature detailing results of studies where ticks were treated with conidial suspension of EPFs under laboratory conditions.

Sr. No.	Entomopathogens (Strains)	Tick Species	Tick Stages	Result	Reference	Country
1	*M. anisopliae* (Ma 319, Ma 359, E9)	*A. cajennense*	Eggs and larvae	EH decreased, EIP and EHP were not different from control	[70]	Brazil
2	*B. bassiana* (Bb 986, Bb 747)	*A. cajennense*				
3	*B. bassiana* (Bb 986)	*A. nitens*	Eggs	EH and control group were same, decreased EHP and EHI	[71]	Brazil
4	*M. anisopliae* (E6S1, E6S2 and CG491)	*B. microplus*	Engorged females	100% mortality of engorged females after 14 days	[72]	Brazil
5	*M. anisopliae* (ATCC 20500) and *B. bassiana* (ATCC 90517)	*Dermacentor variabilis Say*, *I. scapularis Say*, and *R. sanguineus Latrielle*	Nymph Adult	90% nymph mortality, 50–70% adult mortality	[65]	USA
6	*M. anisopliae* (IRAN 437 C, DEMI 001), *B. bassiana* (IRAN 403 C), *L. psalliotae (IRAN 468 C*, *IRAN 518 C*)	*R. (B.) microplus*	Engorged females	90–100% adult mortality 35.5–88% EH reduced	[67]	Iran
7	*B. bassiana* (AM 09, CB 7, JAB 07)	*R. (B.) microplus*	Inoculating eggs, larvae and engorged females	1.36–65.58% EH decreased 0.8–70.49% larval mortality 96–100% adult mortality	[68]	Brazil
8	*M. anisopliae var. anisopliae*	*R. (B.) microplus*	Engorged females	EH decreased 10.69–75.91%	[69]	Brazil
9	*M. anisopliae var. acridum*	*R. (B.) microplus*	Engorged females			
10	*B. bassiana* (Bb28, Bb29 and Bb30)	*R. (B.) microplus*	Egg and larvae	Decreased EH and EHP 30–80%	[34]	Brazil
11	*M. anisopliae* (Ma01, Ma02, Ma04)	*R. (B.) microplus*	Eggs	EH decreased 24% to 83%,	[73]	Brazil
12	*M. anisopliae* (ARSEF3297)	*R. (B.) microplus*	Engorged females	decreased EH, increased EHP	[74]	USA
13	*B. bassiana* (JAB 07, CB 7, AM 9)	*R. sanguineus*	Engorged females	EH decreased	[35]	Brazil
14	*M. anisopliae* (ESC1)	*R. microplus*	Engorged females	100% mortality on 20th day	[75]	Mexico
15	*M. anisopliae* (E9, 319)	*R. (B.) microplus*	Eggs and larvae	EH decreased and EHP	[76]	Brazil
16	*L. psalliotae* (*IRAN 468 C*, *IRAN 518 C*), *B. bassiana* (IRAN 403 C), *M. anisopliae* (IRAN 437 C, DEMI 001)	*B. annulatus*	Engorged females	EH reduced up to 35.5%, 56.3%, and 89.1%, respectively	[67]	Iran
17	*M. anisopliae* Ma14, Ma34	*R. microplus*	Adult and larvae	100% adult mortality on 20th day	[77]	Mexico
18	*M. anisopliae* (M379)	*R. microplus*	Engorged females	55.6% adult mortality on 15th day	[78]	Mexico
19	*Nomuraea rileyi*	*R. microplus*	Engorged females	67.36% adult mortality	[79]	Brazil
20	*B. bassiana* (Bb986)	*R. (B.) microplus*	Engorged females and unfed larvae	71% mortality of larvae on 30th day	[36]	Brazil
21	*M. anisopliae* (Ma 959)	*R. (B.) microplus*		98.7% mortality of larvae on 30th day		
22	*M. anisopliae* (CG 37, CG 384 and IBCB 481)	*R. microplus*	Larvae	100% larval mortality after 14 days	[80]	Brazil
23	*M. anisopliae* (strain 62)	*Hyalomma anatolicum*	Larvae	100% larval mortality after 5 days	[81]	Sudan
24	*B. bassiana* (Bb 112, Bb 113)	*R. microplus*	Larvae	2.5–27% larval mortality on 20th day	[82]	Mexico
25	*I. fumosorosea* (Ifr22)	*R. microplus*	Larvae	28.6% larval mortality on 16th day		
26	*B. bassiana* (CD1123)	*R. sanguineus*	Eggs, larvae, nymph and adult	100% reduction in EH, 100% reduction in larvae to nymph development, 95% adult mortality on 15th day	[37]	Italy
27	*M. anisopliae* (Ma136) *and B. bassiana* (Bb 115)	*R. microplus*	Engorged female	99–100% adult mortality on 15th day	[83]	Mexico
28	*M. anisopliae* (MaV 22, Ma 26, MaV55)	*R. microplus*	Adult ticks	3.3–100% adult mortality on 20th day	[84]	Mexico
29	*M. anisopliae* (TIS-BR030)	*R. microplus*	Larvae	26.3–100% larval mortality in 15 days	[85]	Brazil
30	*M. anisopliae* (MaV 01-54) *B. bassiana* (BbV01-06)	*R. microplus*	Adult ticks	3.3–86.7% adult mortality on 20th day	[86,87]	Mexico
31	*Purpureocillium lilacinum* (PlV01)	*R. microplus*	Adult ticks	94.9% adult mortality on 20th day	[86]	Mexico
32	*M. anisopliae* and *B. bassiana*	*R. sanguineus* and *H. longicornis*	Adult ticks	100% adult mortality in 3 days	[88]	Malaysia
33	*M. anisopliae* (L04, L010, L047, L052, MET 32) *B. bassiana* (LO37)	*I. scapularis*, *D. variabilis*, *R. sanguineus*	Engorged female	100% adult mortality	[89]	Poland
34	*M. anisopliae* (MaV25)	*A. mixtum*	Larvae	32.7% larval mortality on 20th day	[41]	Mexico
35	*Aspergillus flavus* (H-1), *A. nitus* (H-2)	*Haemaphysalis longicornis*	Unfed tick larvae	100% larval mortality at 12th day	[90]	China
36	*M. anisopliae* (CG47)	*R. microplus*	Engorged females	LH reduced	[91]	Brazil
37	*M. anisopliae sensu stricto* (LCM S04)	*R. microplus*	Partially engorged	100% adult mortality in 12 days	[29]	Brazil
38	*M. anisopliae* (LCM S01)	*R. (B.) microplus*	Engorged females	Reduced oviposition, EPI significantly decreased, LH remained same	[92]	Brazil
39	*F. oxysporum*	*R. microplus*	Adults	100% adult mortality	[93]	Pakistan

EIP = egg incubation period; EHP = egg hatchability period; EPI = egg production index; EH = egg hatchability; EHI = egg hatch inhibition; LH = larval hatchability.

**Table 2 insects-15-01017-t002:** Examples from the literature detailing results of studies where ticks were treated with conidial suspension of EPFs under field conditions.

Sr. No.	Entomopathogens (Strains)	Tick Species	Tick Stages	Result	Reference	Country
1	*M. anisopliae* (Ma01, Ma02)	*R. (B.) microplus*	Engorged females	EPI reduced; LH decreased	[94]	Brazil
2	*B. bassiana*	*R. microplus*	Adults	13–38% adult mortality	[95]	South Africa
3	*M. robertsii* (IP 146)	*R. microplus*	Larvae	38.4% larval mortality	[23]	Brazil
4	*M. anisopliae* (LCM S01)	*R. microplus*	Larvae	86% larval mortality	[96]	Brazil
5	*M. anisopliae* (JEF-214, -279, and -290)	*H. longicornis*	Nymphs	80% nymph mortality in 7 days, rising to 100% in 14 days	[97]	Korea
6	*B. bassiana (Baubassil ^®^)*	*R. (B.) microplus*	Adults	84.8% adult mortality	[98]	Colombia
7	*M. anisopliae* (ICIPE 7) and *B. bassiana* (ICIPE 718)	*Rhipicephalus decoloratus*	Larvae	100% larval mortality on 20th day	[99]	Kenya
8	*M. anisopliae* (ESALQ 1037, ESALQ E9)	*R. microplus*	Engorged females	90.53% adult mortality	[100]	Brazil
9	*B. bassiana* (*Balsamo*, *Vuillemin*)	*Hyalomma lusitanicum*	Adults	78.63% reduction of ticks in spring till 30th day, with a 63.28% reduction till 60th day 35.7% reduction in summer till 30th day, with a 29.01% at 60th day	[101]	Spain
10	*M. anisopliae* *sensu lato* X-1c	*I. ricinus*	Larvae and nymphs	92% larval mortality 94% nymphs’ mortality	[102]	Germany
11	*M. anisopliae* (TIS-BR03)	*R. microplus*	Engorged females	97.0% adult mortality	[103]	Brazil
12	*M. brunneum* (strain 7)	*R. (B) annulatus*	Engorged Females	93% adult mortality within 3–4 weeks during summer and 62.2% adult mortality within 6 weeks during summer	[39]	Israel
13	*M. brunneum* (M-7)	*R. sanguineus*	Larvae, nymphs, adults	larval mortality 49%; nymph mortality 79%; adult mortality 25.3%	[104]	Israel
14	*M. brunneum* F52	*I. scapularis*	Nymphs	50% nymph mortality in 7 days	[105]	USA
15	*M. anisopliae* (NA1)	*Rhipicephalus evertsi evertsi* and *R. (B.) decoloratus*	Adult	83% control of tick populations	[106]	Namibia
16	*M. anisopliae* (Ma14, Ma34)	*R. microplus*	Adult and larvae	67–100% adult mortality	[77]	Mexico
17	*M. anisopliae* (Ma 14) *I. fumosorosea*	*R. microplus*	Larvae	94% larval mortality	[107]	Mexico
18	*B. bassiana* (ATCC 74040)	*I. scapularis*	Nymph	38–58.7% control	[108]	USA
19	*M. anisopliae* (F52)			55.6–84.6% control		
20	*M. anisopliae* (Ma34)	*R. (B.) microplus*	Engorged females	45% control at day 1 and 5	[109]	Mexico
21	*M. anisopliae* (ARSEF3297 and IMI386697)	*R. (B.) microplus*	Engorged females	Tick density was reduced (8.5 ± 0.6 and 19.1 ± 0.6 ticks/host) after 3 weeks	[110]	USA
22	*M. anisopliae*	*R. (B.) microplus*	Larvae	86.9% to 94.08% control from day 35 to 48 post-infestation	[111]	Brazil
23	*M. anisopliae* (ESALQ, 959)	*R. (B.) microplus*	Larvae	40.0% control	[112]	Brazil
24	*B. bassiana* (Bb 986)	*A. nitens*	Nymphs	70.1% control	[113]	Brazil
25	*B. bassiana*	*A. nitens*	Adult	50% on day 4–25 after treatment,	[114]	Brazil
26	*M anisopliae* and *B. bassiana*	*A. variegatum* and *R. appendiculatus*	Larvae, nymph, adult	larvae 100%, nymph 40–50%, and adult 80–90%	[66]	Namibia
27	*M. anisopliae*	*R. (B.) microplus*	Engorged females	30% control on reproductive index	[115]	Brazil
28	*M. anisopliae*	*R. (B.) microplus*	Engorged female	43.3% control	[116]	Brazil
29	*M. anisopliae*	*R. (B.) microplus*	Engorged females	52% reduction in EPI	[117]	Brazil

EPI = egg production index; LH = larval hatchability.

## 4. Mode of Action of EPFs Against Ticks

As shown in Table 1 and Table 2, EPFs often exhibit encouraging efficacy against ticks. Given this, it is perhaps surprising that there is a lack of research examining the defense mechanisms employed by ticks during fungal infections, or the mechanisms utilized by these fungi to infect ticks. EPF-based management of plant-feeding invertebrates is already widespread, with numerous examples of these fungi also controlling ectoparasitic species supporting similar potential impact in blood-feeding pests. However, the discussion below highlights the necessity for a more comprehensive understanding of these relationships to make best use of EPFs in tick control, framed against the following key steps in the tick EPFs infection process, as proposed by Beys-da-Silva et al. [118], and as also displayed in Figure 1.

Recognition of the susceptible host by EPFs;Adhesion of EPF conidia to the tick’s cuticle and subsequent germination;Development of specialized EPFs structures, including germ tubes and appressoria;Penetration of EPFs through the tick’s cuticle;Vigorous EPF growth within the tick, leading to the death of the host;Production of conidia once the EPF hyphae emerge through the tick’s cuticle.

### 4.1. EPFs Host Recognition, Conidial Adhesion, and Germination on the Host Cuticle

Air-borne conidia of EPFs land on the surface of the host’s cuticle, which is facilitated by hydrophobic mechanisms [119]. This adhesion process is primarily mediated by surface proteins known as adhesins [25] and hydrophobins [120], with the former having been identified in *B. bassiana*, also demonstrating the existence of homologous proteins [121]. The lipolytic activity observed in ticks enhances this process, specifically through enzymes including lipase and esterase, which have been identified as important in the adhesion and recognition of conidia during the process of infection in ticks by *Metarhizium* [122,123].

### 4.2. EPFS Penetration of the Host Cuticle

Once the conidia have firmly attached to a host, they enter a germination phase, facilitated by favorable environmental conditions. This germination process gives rise to a germination tube, which is followed by the formation of an appressorium or penetration peg that enables EPFs penetration into the tick’s cuticle [124]. Penetration is an active process that depends on the coordinated activity of hydrolytic cuticular enzymes like chitinases and lipases, as well as proteases, in addition to the physical force applied by the penetration peg or appressorium [124]. Various layers of cuticle contain different types of polymeric substrates which are degraded by the activity of the above enzymes, viz. proteases and chitinases [122]. Several proteases, including chymotrypsins, metallopeptidases, trypsins, exopeptidases, subtilisins, and aspartyl peptidases are involved in this process [125]. The expression of these proteases by EPFs is influenced by the specific composition of the host tick cuticle [126], where species of *Metarhizium* and *Beauveria* can produce as many as 11 different subtilisins, with the Pr1 subtilisin-like peptidases being particularly important for the pathogenicity of these EPFs against arthropods. These enzymes may also deliver further beneficial functions, playing a crucial role in both cuticle hydrolysis and nutrient acquisition for EPFs [121,125].

### 4.3. Development of EPFs and Release of Toxins in Tick Body

Upon entry into the host, the EPF develops to produce both mycelium and spores. These structures spread in the whole body of the tick by multiplication, utilizing its circulation system to invade various tissues. This colonization process serves as a pathway for nutrient absorption and establishment within the host [124,127], with virulence factors employed during this stage contributing to the spread of the EPFs within the arthropod’s body, eventually leading to the host’s death. Notably, mycotoxins produced by various EPF species during their growth, including *Beauvericin*, *Beauverolides*, *Bassiannolide*, and *Destruxins*, play a crucial role as toxic substances targeting ticks [127]. These toxins have the ability to disrupt various functions and structures within the tick’s body, including cellular processes, flaccid paralysis, Malpighian tubes, muscular tissues, and the middle intestine [58]. *Beauvericin*, a cyclic hexadepsipeptide belonging to the Enniatin (antibiotic) family, has been particularly well studied and is known to be present in various fungal species, including *B. bassiana* and *Fusarium* [128,129]. Apart from demonstrating insecticidal efficacy, this toxin also exhibits antiviral, antibacterial, and antifungal activity, being used alongside ketoconazole or miconazole for fungal control [130] and even showing promise against cancer or viral and bacterial infections in humans [131].

Once the host fully succumbs to the infection and the EPFs’ nutrient resources are depleted, the fungus breaks through the host’s outer covering, forms aerial mycelia, and begins the process of sporulation on the cadaver to aid in dispersion of the fungal biomass to infect a new host [25]. EPFs possess pathogenicity or virulence factors that confer them with the ability to target their hosts (e.g., ticks) during this phase, potentially allowing for effective pest control while minimizing harm to beneficial non-target organisms [65]. Nevertheless, EPFs are commonly regarded to lack specificity within certain host families, such as mites, and the extent of their specificity towards ticks has not been conclusively demonstrated [24]. Similarly, whilst specificity has been documented in numerous insect species, including *Lymantria dispar*, *Diprion pini*, *Dendrolimus pini*, *Dendrolimus punctatus*, *Malacosoma disstria*, and *Fiorinia externa* [132], several studies have suggested that even where EPFs specialize in a given species they may retain their capability to infect a wider range of hosts [118]. Within the *Hypocreales* order, for example, there are frequent observations of host range variations occurring at both species and strain levels, which have been attributed to the influence of environmental factors on EPF virulence characteristics [133]. Additionally, the pathogenic mechanisms of EPFs can also be affected by the host’s immune response and biological traits. A recent study provides evidence for the importance of EPF strain selection, by demonstrating significantly higher mortality rates in *R. microplus* and *Amblyomma mixtum* ticks after being exposed to a specific strain of EPFs. The experiment involved treating the ticks across four consecutive cycles, where EPF was used as a biological control agent [18]. Although the molecular and metabolic mechanisms responsible for the fungi’s increased effectiveness are not yet fully understood, the results clearly indicate that EPFs can effectively kill these tick species, especially when the most efficacious strains are used.

### 4.4. Ovicidal Effects of EPFs on Ticks

Entomopathogenic fungi are known for their ability to infect the eggs of various tick/insect species [134]. One such specific fungus is *Purpureocillium lilacinum*, which has been found to infect tick eggs and impede the emergence of larvae from them, with the fungus producing enzymes that break down the protective eggshell [135]. Notably, serine protease enzymes secreted by *P. lilacinum* play a key role in causing structural changes to the tick eggshell. Across several studies, the efficacy of certain isolates of *P. lilacinum* in significantly hindering the development of tick eggs has been demonstrated [28].

Aside from the enzymatic route, EPFs employ various other mechanisms to parasitize tick eggs. Fungal spores may attach to the surface of tick eggs and penetrate the eggshell, for example, leading to the death of the developing embryos directly [28,136], or disruption of egg development that interferes with the normal development of tick embryos. This can result in the emergence of abnormal or weakened larvae that are less likely to survive, reduced hatchability, and/or delayed hatching of larvae from eggs, thus impacting tick population growth [137]. Importantly, EPFs can persist in the environment and act as a reservoir, infecting ticks and their eggs over time.

Moreover, EPF may even offer the potential to reduce the burden of TBDs by directly targeting the pathogens that the ticks harbor, reducing the vector load within the ticks themselves to limit infection. Few studies seem to have explored this possibility in ticks, though work with mosquitoes suggests that treatment with *B. bassiana* did not have an influence on levels of malaria parasites within *Anopheles stephensi* [138]. This perhaps suggests that a more promising role for EPF is to target the vector directly.

### 4.5. Impact of EPF on Tick’s Immune System

It has been confirmed that the tick immune system is able to respond to infection by EPF. In work by Fiorotti *et al*. [139], for example, *I. ricinus* hemocytes were capable of phagocytosing *Metarhizium robertsii* conidia, protecting ticks against mild infection with this EPF. Studies exploring tick immune responses to EPF are nevertheless limited, and more work in this area could be recommended.

## 5. Factors Influencing the Effectiveness of EPFs in Tick Control

The effectiveness of EPFs is influenced by a wide range of factors. These factors include both biotic factors, like host range, latency, spore density, and dispersal, along with abiotic factors such as soil properties, temperature, humidity, rainfall, and sunlight. These factors dictate two distinct host ranges of EPFs, i.e., the ecological and physiological host ranges. The ecological host range refers to the infection and survival of EPFs on different host species under natural environmental conditions, whereas the physiological host range is defined by the EPFs’ infection and survival on various species under controlled conditions. Whilst the latter can be confirmed by experimentation under laboratory conditions, exploring the ecological host range poses substantial challenges and remains an area with limited research [133].

### 5.1. Ticks’ Counter Defenses

Several host-related factors, including the presence of a dark epidermal surface, high host density, and immunity, can influence the infection process of EPFs on ticks [140]. As would be expected, ticks have evolved defense mechanisms to counter fungal infections, which include increasing the production of antifungal compounds and activating innate immune responses, including reactive oxygen molecules, humoral melanization, and phagocytosis. As a result, EPFs intended for use as biocontrol agents must possess mechanisms to effectively overcome these host immune barriers and defense adaptations. Such mechanisms have, however, been observed in *B. bassiana*, which demonstrates increased levels of superoxide dismutase expression. This enzyme can play a crucial role in countering oxidative stress through detoxification of the extra hydrogen peroxide into water and molecular oxygen, thereby bolstering the fungus’ capacity to endure such conditions [141].

### 5.2. Environmental Factors

The EPF infection process, particularly the interactions between hosts and fungi, is significantly influenced by abiotic factors in the natural environment [142]. Environmental factors such as soil acidity, soil texture, and the abundance of organic matter, can all have a notable impact on the presence of EPFs, contributing significantly to colonization rates of these organisms on their hosts [140]. Temperature and humidity are also important environmental factors, with high RH levels typically needed to sustain the sporulation of EPFs [143].

Typically, EPFs display optimal spore germination and conidia viability at relative humidities as high as 95.5% and over. However, both *Beauveria* spp. and *Metarhizium* spp. have demonstrated the ability to infect their respective hosts even in conditions of low atmospheric humidity within microhabitats [140]. Studies have also consistently revealed that *B. bassiana* and *M. anisopliae* isolates exhibit strong growth across a broad temperature range, spanning from 8 to 37 °C. However, according to Abdul Qayyum et al. [140], the most suitable temperature for the germination and growth of EPFs consistently falls between 20 °C and 30 °C, with strains originating from warmer regions tending to perform better at higher temperatures, and vice versa [140].

Unlike *B. bassiana*, *M. anisopliae* demonstrates relatively strong tolerance to UV-B radiation. Furthermore, research has unveiled that natural isolates of both *Beauveria* spp. and *Metarhizium* spp. surpass non-native strains in terms of solar exposure tolerance and virulence, demonstrating adaption to local conditions from which they were originally isolated [37,144]. This may be an important consideration when deploying commercially formulated EPFs into environments that differ significantly from those found in the culturing/manufacturing facility for the product.

Soil acidity can also be important to EPF development and survival, where both *B. bassiana* and *Metarhizium* spp. display peak growth within a pH range of 3 to 9. In particular, mycelium growth thrives at pH 4.4, while pH 6 is ideal for optimal spore production [145]. Interestingly, while these fungi do not require a specific nitrogen source, urea has been found to be highly effective in promoting sporulation, particularly at higher concentrations.

Being living organisms, EPF will perform best within their preferred environmental ranges, with the risk that local climate change effects may render some EPF species unsuited to geographic areas where the limits of these ranges are exceeded. Whilst developments in product formulation technology may help to infer aspects of increased EPF climate resilience, a recent review emphasizes the importance of selection and use of “environmentally competent” strains to mitigate this risk.

## 6. Challenges and Limitations of Using EPFs Against Ticks

There are a number of issues and restrictions with using EPFs to control ticks that need to be resolved [18]. For one, current tick control methods, such as synthetic acaricides that affect the physiology, reproduction, or survival of ticks, might also exert negative effects on EPFs. Such incompatibility of conventional controls with EPFs has been demonstrated in other Acari and dictates that care must be taken when including EPFs within integrated tick management programs.

The effectiveness of EPFs can depend on the type of tick and its stage of development, where whilst some fungi show increased potency against specific species of ticks, others have little to no effect [24]. Interactions between culturing conditions and the genetic/phenotypic characteristics of the EPFs can also be important here, where Iwanicki et al. [146] concluded that two particular strains of *Metarhizium* (ESALQ1426 and ESALQ4676) delivered notably high efficacy against arthropods, likely due to their enhanced production of blastospores under the culturing conditions used (increased glucose and accelerated fermentation using pre-cultured, yeast-like cells). This further demonstrates a role for optimizing culturing techniques and EPF manufacturing protocols, where the authors also explored air-dried blastospores efficacy against *R. microplus* larvae and obtained good results. The importance of strain selection, as also noted earlier, has been confirmed in multiple works considering the worldwide management of ticks through the utilization of various disease-causing agents [62,63], where the value of identifying and culturing optimal fungal strains from the rich global diversity of EPF species is noted.

The environment in which EPFss are used is extremely important to their success [147] where, as previously noted, these fungi are extremely sensitive to heat, humidity, and ultraviolet light (UV-A and UV-B). Environmental parameters that are sub-optimal prevent fungal growth and reduce EPFs capacity to successfully infect and kill ticks [148]. It can, however, be difficult to ensure ideal conditions throughout the entire EPF application process, especially in outdoor settings where ticks are frequently found. In addition, arthropod hosts have also evolved mechanisms to resist infestation by EPFs, with many producing metabolites that lower the efficacy of EPFs [149,150].

The accessibility and availability of EPFs present another restriction to their use. Although there are many commercially available products containing EPF species, there are comparatively few fungal strains and formulations created specifically for controlling ticks [23,151], perhaps reflecting the difficulties in overcoming challenges regarding EPFs’ longevity and persistence in the outdoor environment in which EPFs would need to perform to target these pests. Conversely, relatively more EPF pesticides have been developed for use in controlled conditions (e.g., in glasshouse crops where environmental parameters are carefully managed), where efficacy against pests is also typically achieved by implementing a strategy that only needs to rely on the activity of the EPFs at the point of application, rather than relying on the continued development of successive fungal generations [152]. Within this framework, more than 170 EPF products have been created, drawing from a pool of at least 12 fungal species.

Despite there being an estimated 700 EPF species spanning around 90 genera, the commercially cultivated fungi primarily belong to *Beauveria*, *Metarhizium*, *Lecanicillium*, and *Isaria* species, likely due to their ease of large-scale production. Rather than expanding the EPF species used in products, the primary focus of many manufacturers has centered on technical aspects of biopesticide development, encompassing mass production, formulation, and the selection of fast-acting strains. Production prerequisites encompass cost-effectiveness, long-term stability, and, crucially, consistent field efficacy. Common methodologies to achieve these goals include generating dispersal units (diaspores) through inducing aerial conidiation on solid growth substrates, cultivating blastospores through yeast-like growth in liquid media, or cultivating hyphal biomass in either liquid or solid mediums [153,154,155]. Even with optimized manufacturing procedures, however, EPFs normally need to be applied repeatedly to achieve acceptable levels of pest control, which can increase the cost of employing these biocontrol agents, both in terms of buying the product and the extra labor needed for regular application.

Potential safety issues related to the use of EPFs must also be considered. Although these fungi are typically thought to be harmless for people and other non-target organisms, specific safety measures must still be taken to reduce any potential cross-species risks, especially for other invertebrates (see earlier). This entails using the right handling, storage, and application procedures to prevent unintentional exposure to people, animals, or beneficial insects [142,156,157,158].

A final challenge is that, depending on a particular situation and scope of use, the general efficacy of EPFs against ticks may vary. A given product may be efficient in restricted locations (e.g., a confined area), while its efficacy might decline when used in expansive landscapes with a variety of tick populations [159]. With this and the numerous other challenges above in mind, the integration of EPFs into existing tick control programs or strategies requires careful coordination and local adaptation to ensure synergistic effects and maximize overall tick population reduction.

In order to help overcome some of the above challenges, research into how to expand the range of biotic and abiotic parameters under which EPFs can perform well would be particularly welcome, seeking to make EPFs efficacy less dependent on temperature, humidity, and other environmental parameters. Similarly, identifying an EPF strain that could target multiple arthropod pest species, such as ticks, mites, mosquitoes, and flies, without affecting beneficial insects and other organisms would be groundbreaking. The identification of EPF strains displaying resistance to tick counter defenses, such as antifungal molecules, would also be worth pursuing to ensure optimum efficacy and impact. Finally, it would be useful to develop cost-effective EPF release devices that are able to sustain their populations and efficacy over a longer period, allowing single “applications” to target multiple stages and/or generations of the tick life cycle.

## 7. Regulatory Aspects and Guidelines for the Use of EPFs Against Ticks

Due to the growing need for sustainable and eco-friendly alternatives to conventional chemical pesticides, regulatory aspects and recommendations for the use of EPFs against ticks have attracted a lot of attention in recent years. In comparison to synthetic pesticides, utilizing EPFs reduces possible dangers to human health and the environment while providing a promising tick control method [160]. Nevertheless, regulatory frameworks and standard application procedures have been established to guarantee the safe and efficient use of these fungi [161]. Similarly to conventional products, numerous policymakers, governmental organizations, academic institutions, and business representatives are involved in the regulatory process controlling the use of EPFs against ticks [162], from efficacy testing and environmental toxicity testing to registration of products for field use [163], also evaluating consumer safety should fungal spores enter the food chain [164].

One of the key considerations in regulatory guidelines is the selection of appropriate fungal strains [165]. As repeatedly noted above, it is essential to identify and characterize the most effective and host-specific EPF strains for tick control [166], and regulatory bodies typically require extensive scientific data on the taxonomy, virulence, and host range at strain level. This information ensures that the fungi target ticks specifically, without posing a significant threat to non-target organisms [167].

Regulatory guidelines also place a strong emphasis on the creation of uniform formulations and application techniques, with standards for composition, stability, and quality control of fungal products provided [162,168,169], alongside details on application methods, doses, and timing to maximize effectiveness and reduce negative environmental effects [142]. For products to make it to market, consistent and repeatable outcomes in field tests are needed, with these regulations also specifying that handling, application, and disposal information must be included on product labels to support safe and effective use [170,171]. Clear and precise labeling has been noted as being essential to achieving this [172], and these recommendations should also stress any necessity for workers to receive the right training and certification prior to deploying a product [160].

Programs for long-term monitoring and surveillance are essential components of any regulatory framework. Here, the efficiency of EPFs is assessed through routine monitoring in treated locations, which also serves to identify any potential negative impacts on non-target organisms or the surrounding ecosystem [173]. Such monitoring programs also aid in identifying emerging resistance in tick populations and developing appropriate strategies to manage it accordingly [174], facilitating the choice of suitable strains, standardized formulations, and secure application techniques [152]. In short, regulatory processes and procedures already exist to support EPF development and deployment against ticks, where following these rules and recommendations encourages the responsible and sustainable use of EPFs, providing an efficient, sustainable, and safe tick control method.

## 8. Conclusions

EPFs represent a specialized group of micro-organisms with significant potential as biological control agents against ticks and other arthropods. As the limitations and negative environmental impacts of chemical acaricides become more evident, the demand for sustainable and eco-friendly tick control solutions has grown. EPFs, including species of the genera *Beauveria*, *Metarhizium*, *Lecanicillium*, and *Isaria*, have demonstrated efficacy in controlling a range of tick species, with numerous examples of their effective use in laboratory and field settings available in the literature, and presented within this review. Developing an improved understanding of the mode of action of EPF will allow researchers to optimize IPM programs for tick control, permitting the selection of strains best suited to deliver safe, high-efficacy control that complements local co-control methods being deployed and minimizes risks to non-target organisms and the wider environment. These fungi employ unique modes of action, infecting and ultimately killing ticks through various mechanisms, where their ability to colonize ticks and persist within their hosts offers a promising avenue for long-term tick management approaches. Research efforts have predominantly focused on understanding the ecological interactions between these fungi and their hosts, including the factors that influence successful colonization, such as inoculation methods and environmental sensitivity.

While commercial applications of EPF are currently limited to a few fungal species, ongoing research is paving the way for more efficient use of existing products as well as the development of new EPF species for commercial use. Ongoing development of mass production techniques and EPF formulations with extended shelf lives could be particularly important for ensuring the accessibility, cost-effectiveness, and residual activity of these products, the latter of which will be especially relevant to ensure efficacy against ticks when deployed in outdoor settings. Development of new EPF strains could also be supported through new and emerging technologies. Advances in molecular sequencing systems, for example, nanopore technologies, are allowing for improved and increasingly affordable/mobile genetic characterization pipelines, which could be used to identify novel EPF strains.

## Figures and Tables

**Figure 1 insects-15-01017-f001:**
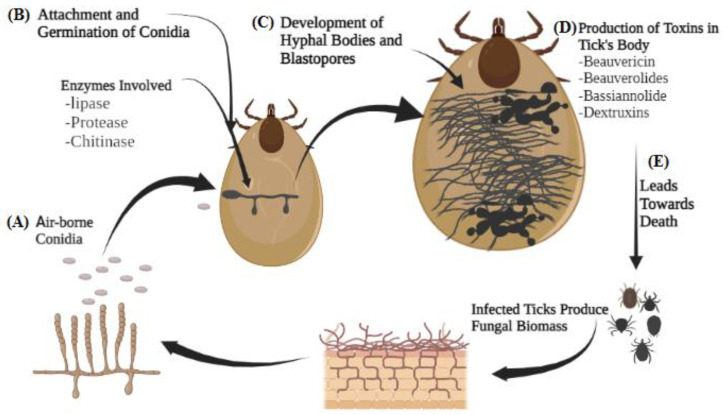
Schematic representation of the mode of action of EPFs against ticks. (**A**) Air-borne EPF conidia land on the cuticle of ticks attracted by hydrophobic mechanisms, (**B**) after adhesion and firm attachment germination and penetration into cuticle of the host takes place in the presence of cuticle degrading enzymes, viz. lipases, proteases, and chitinase. (**C**) After penetration, EPFs produce bunches of mycelia, fungal biomass, and blastospores, (**D**) which produce toxins in the tick’s body. (**E**) These toxins cause destruction of cellular processes, Malpighian tubes, muscular tissues, and middle intestine and insight flaccid paralysis in the tick’s body, which leads to death.

## Data Availability

No new data were created or analyzed in this study. Data sharing is not applicable to this article.

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
