# Peer review of "Entomopathogenic Fungi as Alternatives to Chemical Acaricides: Challenges, Opportunities and Prospects for Sustainable Tick Control"

_insects, 2024, doi:10.3390/insects15121017_

Round 1
Reviewer 1 Report
Comments and Suggestions for Authors
This manuscript addresses a significant and emerging field of study, reflecting a growing interest in the topic. The objective of this comprehensive review is to provide an overview of the existing literature on the potential of EPF in tick control. The authors concentrate on the mode of action of EPF, an analysis of previous field successes and failures, an evaluation of the advantages of EPF, as well as an investigation of potential applications. The topics addressed in this article are particularly pertinent given that EPFs are regarded as environmentally friendly alternatives to chemical pesticides, particularly in the context of tick control. Moreover, they align with the principles of sustainable and One Health practices.
In general, the manuscript is well-written and easy to follow. In addition, the authors may wish to consider the following aspects when preparing a revised version of the manuscript:
Line 65, 66, 102, 242. 318, 319 The species name should be presented in italics.
Line 82-84 It is my contention that such a modification will be more transparent and comprehensible. „The geographical expansion of vector-borne diseases beyond previous ranges is the result of a complex interplay of factors, including climate change affecting both the tick and host habitat, changes in land use, global trade, and animal movements across various geographical settings”
Line 319-321 „…several studies have suggested that even where EPF specialize in a given species they may retain their capability to infect a wider range of hosts. It is therefore pertinent to consider the potential impact of these strains on the wider ecosystem”. To what extent might their use affect other species, such as the honeybee? Has any research been conducted to assess the risks?
Chapter 6. Challenges and limitations of using EPF against ticks.
In light of the limitations imposed on the utilisation of EPF, what are the prospects for the development of an optimal formulation within the next decade? Such a formulation would need to be safe for fauna and demonstrate stability across a range of conditions, some of which are highly unfavourable for fungal growth. Please provide your insights on this matter.
Reviewer 2 Report
Comments and Suggestions for Authors
Dear Authors
While this review is of significant interest to readers, certain sections may cause a disconnect in engagement due to issues with narrative flow. In my opinion, greater care should be taken to ensure a logical and coherent structure throughout the review. Following a clear flow or outline would help improve the organization and clarity of the manuscript. For instance, you could structure the review as follows: first, define what an entomopathogen is, followed by what an entomopathogenic fungus is. Then, explain the mechanism of action of entomopathogenic fungi, followed by their use against ticks. After that, discuss the challenges associated with using entomopathogenic fungi in tick control, including factors such as temperature, UV light, and humidity that affect efficacy, resistance development, and toxicity to non-target organisms. Finally, include a section with general recommendations and future directions. I suggest revisiting the manuscript with this structure in mind to enhance the flow and maintain reader engagement.
1. I would like to kindly suggest reviewing the English language throughout the manuscript, as there are minor issues that may affect clarity and fluency.
For example;
Whilst a number of studies suggest potential of EPF against ticks, this review suggest that limitations to their effective use may include factors such as heat, humidity, and ultraviolet light (UV-A and UV-B).
………this review suggests that
2. "Acaricides" could refer to both chemical and non-chemical control methods. To be more specific and highlight the focus on reducing chemical inputs, I recommend revising the title to: "Entomopathogenic Fungi as Alternatives to Chemical Acaricides: Challenges, Opportunities and Prospects for Sustainable Tick Control."
3. Lines 28 and 226: ‘09’ should be revised as ‘9’ sentence:
Overall, 9 different EPF species were used against 15 different species of ticks.
4. Please ensure that all species names are italicized the first time they are mentioned in the manuscript, with the author(s) and year of description included in parentheses immediately after the species name (without italics). For example, the first mention should be written as Metarhizium anisopliae (Metschnikoff) Sorokin, 1883, and in later references, M. anisopliae can be used. Lines 129-130 etc.
Line 55-56; I. scapularis, R. Microplus, I. Ricinus, I. Scapularis and others, Line 318; Lymantria dispar, Diprion pini, Dendrolimus pini, D. punctatus, Malacosoma disstria, and Fiorinia externa
And also please ensure that all genus names, such as Amblyomma, Ixodes are italicized throughout the manuscript.
5. Keywords should be listed in alphabetical order. Additionally, I recommend ensuring that alternative keywords are used instead of simply repeating terms from the title, to increase the discoverability of the article.
6. Line 106; ‘EPF (entomopathogenic fungi)’ should be revised as ‘Entomopathogenic fungi (EPF)’
7. Lines123-124; ‘According to 123 [28], 17 fungal species’ should be revised as ‘According to White et al. [28], 17 fungal species’
8. ‘sensu lato’ should be shortened as ‘s.l.’ in text.
9. Line 177: What is the (REF)? Check all text?
10. Lines 207-209: ‘Entomopathogenic fungi’ revised as ‘EPF’
11. Line 221: ‘lifecycle stages’ should be revised as ‘life stages’
12. Line 242: ‘Anocentor nitens’ should be italic
13. In Figure 1, please ensure that all stages are numbered directly in the figure for clarity. Additionally, the corresponding explanations should be provided in the figure caption.
14. Line 429; ‘he importance’ should be revised as ‘The importance’
15. Lines 531-544: I suggest removing the section under the heading "Other nature-based solutions to tick control," as the primary focus of this review is on entomopathogenic nematodes. Including this section might divert attention from the main topic and reduce the coherence of the manuscript.
16. Please ensure that all references are formatted according to the journal’s style guidelines. There are several errors in the current reference list, including incorrect journal abbreviations, missing italics for species names, and other formatting issues.
Comments on the Quality of English LanguageIt must be checked for grammer and spelling.
Round 2
Reviewer 2 Report
Comments and Suggestions for Authors
Line 79: Genus names such as Ixodes should be italicized.
The period after the word "regions" in line 88 should be removed
Species names and genus names used to define species within the text should be written out fully upon their first mention; thereafter, abbreviations should be used.
Comments on the Quality of English LanguageGood
